# Persulfidation of DJ-1: Mechanism and Consequences

**DOI:** 10.3390/biom13010027

**Published:** 2022-12-22

**Authors:** Erwan Galardon, Nicolas Mathas, Dominique Padovani, Laurent Le Corre, Gabrielle Poncet, Julien Dairou

**Affiliations:** Laboratoire de Chimie et de Biochimie Pharmacologiques et Toxicologiques, CNRS, Université Paris Cité, F-75006 Paris, France

**Keywords:** persulfide, DJ-1, hydrogen sulfide, sulfinic

## Abstract

DJ-1 (also called PARK7) is a ubiquitously expressed protein involved in the etiology of Parkinson disease and cancers. At least one of its three cysteine residues is functionally essential, and its oxidation state determines the specific function of the enzyme. DJ-1 was recently reported to be persulfidated in mammalian cell lines, but the implications of this post-translational modification have not yet been analyzed. Here, we report that recombinant DJ-1 is reversibly persulfidated at cysteine 106 by reaction with various sulfane donors and subsequently inhibited. Strikingly, this reaction is orders of magnitude faster than C106 oxidation by H_2_O_2_, and persulfidated DJ-1 behaves differently than sulfinylated DJ-1. Both these PTMs most likely play a dedicated role in DJ-1 signaling or protective pathways.

## 1. Introduction

DJ-1 (also called PARK7) is a small (~20 kDa), ubiquitously expressed homodimeric protein. Since the report that mutation of its encoding gene in humans leads to autosomal recessive early-onset Parkinson disease (PD) 7 [1], intensive studies have been undertaken to decipher its function and its role in the etiology of this neurodegenerative disease. Thus, DJ-1 has been proposed to take part in various physiological pathways related to the promotion of cell survival [2]. For instance, DJ-1 activates the extracellular signal-regulated kinase pathway ERK1/2 [3] and the phosphatidylinositol-3-kinase (PI3K/Akt) pathway [4]. It also modulates oxidative and electrophilic stresses. For example, DJ-1 activates the Nrf2-mediated antioxidant response [5], catalytically protects various biomolecules against glycation by methylglyoxal [6] or detoxifies reactive compounds produced during glycolysis [7,8], although the nature of its physiological substrates is still a matter of controversy [9,10]. In addition to PD, DJ-1 is now proposed to be involved in various pathological settings, such as ischemia–reperfusion injury [11], inflammatory bowel disease [12], diabetes [13] or cancers [14].

Human DJ-1 is characterized by an α/β-flavodoxin fold core, and it possesses three cysteines (C46, C53 and C106), the latter being highly conserved and localized in the nucleophile elbow region. C106 has been identified as an important residue whose thiolate group is key to most of DJ-1 functions [5], its oxidation state determining the specific function of the enzyme. Thus, under a reduced catalytic C106 status (CysS^−^) [5], DJ-1 has been proposed to act as a peroxiredoxin-like peroxidase [15], a protease [16], a glyoxalase [8], a deglycase [17] and more recently as a scavenger of a reactive glycolytic metabolite [7]. Additionally, DJ-1 displays a non-physiologically relevant esterase activity which was used to develop an assay to screen for new DJ-1 inhibitors [18]. In addition, C106 is characterized by a low thiol p*K_a_* value ~5 [19] and can be oxidized to sulfinate (-SO_2_^−^) and sulfonate (-SO_3_^−^) by hydrogen peroxide (H_2_O_2_) [20,21]. For instance, the sulfinylation of C106 shifts its isoelectric point and promotes its intra-cellular relocation, allowing DJ-1 to play a role in redox sensing and cytoprotection [22,23]. The same post-translational modification (PTM) regulates its participation in the composition of high molecular weight complexes that play a role in RNA metabolism and catecholamine homeostasis in cultured cells and human brain [24,25]. Interestingly, while C106 was clearly demonstrated to be the key cysteine in the aforementioned studies, the sulfinylation of C46 may also have physiological significance as the protein thus modified is one of the few substrates of sulfiredoxin [26].

In addition to being implicated into the various S-oxygenation reactions briefly described above, C106 is also the target of other PTMs. For instance, all three Cys residues of DJ-1 were reported to be nitrosylated (formation of CysS-NO) in various cell lines in conflicting studies [27,28], with C106 seemingly playing a role in trans-nitrosylation processes [28,29]. In addition, C106 was also found to experience persulfidation (formation of CysS-SH), which may prevent it from undergoing uncontrolled S-oxygenation under oxidative stress conditions in MEF cells [30]. In addition to the importance of C106 as a redox sensor, part of the activity of DJ-1 depends on the redox-sensitive removal of a 15-amino acid peptide at its C terminus (C_ter_) [16,31]. For instance, the C_ter_ cleavage of DJ-1 in response to acute myocardial ischemia-reperfusion injury protects from heart failure by inducing anti-glycation properties [32]. However, the nature of the stimulus and the mechanism by which cleavage of the C_ter_ peptide occurs are still elusive. In this context, understanding the reactivity and the effects of various redox messengers on DJ-1 might provide new insights into DJ-1’s role in various cellular contexts and could identify novel mechanisms involved in diseases’ setting and etiology.

Here, we report our first results on the persulfidation of DJ-1 and its outcome on the enzymatic activities and structure of the protein. We show that recombinant human DJ-1 is persulfidated at C106 and inhibited in vitro by reaction with various sulfane sulfur [33] donors, a reaction orders of magnitude faster than sulfinylation. Additionally, recombinant DJ-1 is endogenously persulfidated when overexpressed in *E. coli*. Although persulfidation and sulfinylation both result in DJ-1 inhibition, they lead to proteins with different behavior. These observations suggest different fates for each of these PTMs.

## 2. Materials and Methods

### 2.1. Materials

Most chemical and biochemical reactants were purchased from Merck, Darmstadt, Germany. Sodium di- and tetrasulfide were purchased from Dojindo Molecular Technologies Inc., Rockville, MD, USA and sodium hydrosulfide from Strem, Bischheim, France. These salts were manipulated under argon atmosphere (<1 ppm O_2_) in a glovebox. DAz-2:Cy5 was synthesized as previously described [30]. Recombinant human thioredoxin 1 (hTrx) was purchased from ThermoFisher, Illkirch, France. The plasmid pET-TRSter for heterologous expression of human thioredoxin reductase (hTrxR) was purchased from Addgene, Watertown, MA, USA. Plasmids for wt and mutant DJ-1 were obtained from Dr. Sun-Sin Cha [34]. The plasmid containing the human CSE gene (pET-28-based expression vector incorporating a tobacco etch virus (TEV)-cleavable N-terminal His tag fusion) was a kind gift from Dr. Tobias Karlberg (Structural Genomics Consortium, Karolinska Institute, Stokholm, Sweden). Reactions were typically run in phosphate buffered saline (PBS) containing 200 µM diethylenetriaminepentaacetic acid (DTPA), unless otherwise stated. The buffer was roughly de-gassed by bubbling argon for 30 min before experiments with the sodium hydrosulfide or polysulfide salts. UV-visible spectra were recorded on Cary 300 (Agilent, Santa Clara, CA, USA), Jasco V-700 (Jasco, Lisses, France) or Biotek PowerWave XS(Agilent, Santa Clara, CA, USA) spectrometers. Differential scanning fluorimetry (DSF) experiments were carried out on a Bio-rad CFX96 Real Time PCR system (Bio-rad, Marnes-la-Coquette, France). Gels were imaged on an LAS 4000 (Cytiva, Velizy-Villacoublay, France) or Bio-Rad GelDoc Go (Bio-rad, Marnes-la-Coquette, France), and images were treated with FiJi (https://imagej.net/software/fiji/downloads (accessed on 8 November 2022)). The liquid chromatography coupled to mass spectrometry (LC-MS) system was composed of a Shimadzu apparatus equipped with a LC30AD pump and a kinetex 5u C18 100A column, a SiL30AC auto-sampler coupled with a photodiode array detector PDA20A and a triple quadrupole mass detector 8060 (Shimadzu, Noisiel, France). Fitting of the data was performed with SigmaPlot 10 (Systat Software, San Jose, CA, USA). Statistical analysis was carried out using the Excel (Microsoft) data analysis package: Each set of activitiesor persulfidation levels were compared with the relevant control using unpaired two-tailed *t*-test.

### 2.2. Proteins Expression and Purification

Proteins were expressed and purified as previously described [17], with the exception that the last purification step of DJ-1, i.e., hydroxyapatite column, was carried out using PBS buffer without chelator or dithiothreitol (DTT). Cystathionine γ-lyase (CSE) was expressed and purified following the described procedure [35]. hTrxR was overexpressed in BL21(DE3) cells and purified as follows: *E. coli* cells were re-suspended in 100 mL extraction buffer (50 mM Tris-HCl, pH 7.5, 30 mM KCl, 5 mM DTT, 1 mM EDTA, 1 mM pMSF and 2 tablets of cOmplete^TM^ protease inhibitor cocktail) and sonicated on ice for 5 min (10” on; 40” off). Following centrifugation for 60 min at 20,000 rpm and 4 °C, the supernatant fraction was submitted at first to a streptomycin sulfate (2.5 % *w*/*v*) precipitation step, then treated with pancreatic DNAse I (3 mg) after centrifugation and lastly precipitated with saturated ammonium sulfate (30–85 %) for 60 min at 4 °C. The solution was centrifuged as described above, and the yellow pellet was re-suspended in 50 mM Tris-HCl, pH 7.5, 1 mM EDTA (buffer A) and dialyzed overnight against the same buffer at 4 °C. The protein extracts were then loaded on a HiPrep DEAE FF 16/10 column (Cytiva). hTrxR was eluted with a linear gradient from 10 to 500 m M KCl in buffer A. Fractions containing hTrxR, as judged by SDS-PAGE 12%, were pooled, concentrated with a centricon YM10 and applied to a HiLoad Superdex 200 (Cytiva) equilibrated with buffer A complemented with 100 mM KCl. The enzyme was submitted to an isocratic elution, and fractions containing pure hTrxR were dialyzed against 50 mM potassium phosphate, pH 7.5, 1 mM EDTA and stored frozen at −80 °C.

### 2.3. Reactions between DJ-1 and 2,2-Dithiodipyridine

To a 180 µM solution of 2,2-dithiopyridine in the suitable buffer was added an 18 µM solution of DJ-1 or its mutant C106S in the same buffer and the absorbance at 343 nm recorded over time. Absorption time profiles were fitted with SigmaPlot 10 to the double exponential A1[1−exp(−*k*_obs1_ t)] + A2[1−exp(−*k*_obs2_ t)] + B or the mono exponential A[1−exp(−*k_obs_* t)] + B function.

### 2.4. Reactions between DJ-1 and Sulfane Sulfur Donors

DJ-1 (12.5 µM) was incubated with various donors at room temperature (RT) for 30 or 60 min, and its esterase activity was determined by monitoring the slope of the absorbance at 405 nm vs. time at 25 °C upon addition of 10 µL of the aforementioned solution to 190 µL of a 2.8 mM solution of *p*-nitrophenylacetate (*p*NPA, prepared in PBS without DTPA from a 200 mM stock solution in dimethylsulfoxyde (DMSO)). This activity was compared to the activity of a similar DJ-1 solution without donor (100%). Solutions of the donors without DJ-1 were also used as control and show no significant esterase activity unless otherwise stated. Kinetic data with Na_2_S_2_ were obtained at 25 °C as follows: to 5 mL of a 0.625 µM solution of DJ-1 was added 1.25, 2.50, 5.00 or 7.50 µL of a 10 mM solution of Na_2_S_2_ (final concentrations: 2.5, 5.0, 10.0 and 15.0 µM). The esterase activity was recorded at various time intervals by monitoring the slope of the absorbance at 405 nm vs. time upon addition of 1 mL of the aforementioned solution to 3 µL of a 200 mM solution of *p*NPA in DMSO. The pseudo-first order rate constants *k*_obs_ were obtained for each concentration by fitting the data with SigmaPlot 10 to the mono exponential function A[1−exp(−*k*_obs_ t)] + B. The bimolecular rate constant *k*_inac_ is the slope of the linear fit obtained from the plot of *k*_obs_ vs. [Na_2_S_2_].

DJ-1 protection of guanosine monophosphate (GMP) against modification induced by methylglyoxal (MGO) was determined by LC-MS. GMP (40 µM) was incubated with MGO (400 µM) and either wt DJ-1 (4 µM) or wt DJ-1 + Na_2_S_2_ (4 µM, from a stock solution prepared by incubating 40 µM DJ-1 with 60 µM Na_2_S_2_ for 20 min at 20 °C) for 12 min at 37 °C. Data were calculated by measuring the surface area of the glycated GMP peak (m/z = 436.09, ESI+) or lactate (m/z = 89.00, ESI−). Note, 100% protection corresponds to the reaction [GMP + MGO + DJ-1], while 0% protection corresponds to the reaction [GMP + MGO], since 6 µM Na_2_S_2_ did not influence the reaction between GMP and MGO.

### 2.5. Reactivation of Persulfidated DJ-1

A 13 µM solution of DJ-1 was inhibited by reaction with 15 µM of Na_2_S_2_, leading to an 80% drop of its esterase activity (see below). This solution was then incubated with either 1 mM DTT, the system hTrx/hTrxR/NADPH (8 µM/2.5 µM/400 µM) or the system hTrx/DTT (8 µM/1 mM) and the esterase activity recorded as described above at various time points.

### 2.6. Differential Scanning Fluorimetry Experiments

Solutions of DJ-1 (50 µM) in PBS were incubated with/without 65 µM Na_2_S_2_ or 350 µM H_2_O_2_ for 20 min at RT, then de-salted with Micro Bio-Spin 6 columns (Bio-Rad). To 22.5 µL of the resulting solutions was added KCl (final concentration 100 mM) and 2.5 µL of a 50× stock solution of Sypro orange in water. DSF assays were run using qPCR plates using the following parameters: samples were heated from 10 to 95 °C at a rate of 2 °C/min, and fluorescence was recorded using the FRET channel. It is noteworthy that DTT (1 mM) was added to the sulfinylated form of DJ-1 prior to qPCR as its omission precluded the recording of a clean melting curve. Data were plotted as the first derivative of fluorescence as a function of temperature, whose peak corresponds to the melting temperature (Tm). Identical Tm values were obtained by running the assay after mixing 20 µL of a 50 µM solution of DJ-1 in PBS without DTPA with 2.5 µL of a 650 µM solution of Na_2_S_2_ and 2.5 µL of a 50× stock solution of Sypro orange.

### 2.7. In Gel Detection of Persulfidation

(i) With purified proteins: wt or mutant (C106S) DJ-1 (approx. 1.5 mg/mL in PBS, 75 µM) were first treated with/without sodium disulfide (100 or 150 µM final concentration) for 20 min at RT, then sodium dodecyl sulfate (SDS) was added to reach a 2% final concentration. The resulting mixtures were then treated at 37 °C with 20 mM NBD-Cl for 1 h. Proteins were then precipitated with water/methanol/chloroform (4/4/1), and the resulting pellet was re-suspended in Hepes buffer (50 mM, pH = 7.4) containing 2% SDS and further incubated at 37 °C with Daz-2:Cy5 mix (50 µM) for 30 min [30]. After precipitation and re-suspension of the pellet as above, the solutions were submitted to a de-naturing, non-reducing 12% criterion XT Bis-Tris gel (Bio-Rad).

(ii) With *E. coli* extracts: The plasmids for wt or mutant C106S DJ-1 were transformed into BL21(DE3) *E. coli* strain, and the bacteria were grown overnight in LB medium containing 100 µg/mL ampicillin. Next, the overnight culture was added at 2% *v*/*v* to LB medium supplemented with ampicillin (100 µg/mL). Bacteria were grown at 37 °C until OD600 = 0.6, and IPTG (0.2 mM final) was added to induce protein expression. If needed, thiosulfate (25 mM) or cysteine trisulfide (1 mM) [36] was added 30 min after induction. After 5 h culture at 37 °C under mild shaking (150 rpm), bacteria were pelleted, re-suspended in PBS buffer supplemented with cOmplete^TM^ protease inhibitor cocktail (Roche) and 4-Chloro-7-nitro-2,1,3-benzoxadiazole (NBD-Cl, 25 mM) and sonicated twice (20” on, 60” off) on ice. After 1 h at 37 °C, the resulting extracts were centrifuged at 20,000 rpm for 20 min, then the proteins precipitated using methanol and chloroform and further treated as previously described [30], before loading on a de-naturing, non-reducing 12% criterion XT Bis-Tris gel (Bio-Rad).

### 2.8. Molecular Dynamics

The crystal structure of the DJ-1 dimer was downloaded from the Protein Data Bank (PDB ID: 3SF8, 05/10/2011, https://www.pdb.org (accessed on 8 November 2022) [37]). The dimer formed by sub-units A and B was prepared using the Prepare Protein module in Biovia Discovery Studio^®^ (DS) 2021 with the default parameters and the CHARMM force field. Briefly, all crystallographic water molecules were removed. Bond orders were assigned, and hydrogen and missing atoms were added. The protonation states on protein were adjusted at pH 7.4. C106 was deprotonated and E18 protonated on the basis of previous studies [19], sulfur or oxygen atoms were added to the sulfonate form of C106 when required, and the structures were minimized.

To assess the influence of persulfidation at C106 and compare it to the wt or sulfinylated protein, 70 ns molecular dynamics (MD) simulations were run using the CHARMM36m force field and the NAMD protocol [38] implemented in DS 2021. Proteins were solvated in a cubic box using a TIP3P water model. Periodic boundary conditions were applied with a minimum distance of 10 Å from periodic boundary, and Na+Cl- counter ions were added to neutralize the system. Solvated complexes were subjected to 2 cycles of energy minimization (1000 steps of Steepest Descent algorithm, then 20,000 steps of ABNR algorithm) followed by 500 ps of heating from 50 to 300 K at constant volume, 1 ns of equilibration at 300 K and 50 ps of production in the NPT ensemble (300 K, 1 atm). All MD simulations were performed under NPT conditions (300 K, 1 atm). Langevin Dynamics and Langevin Piston methods were applied to control the temperature and the pressure. Short-range electrostatic and Van der Waals interactions were computed with a 12 Å cut-off distance, and long-range electrostatic interactions were treated by the Particle Mesh Ewald (PME) method. All bonds with hydrogen atoms were held rigid using the SETTLE algorithm. RMSD and RMSF values, as well as distance and interface energies, were calculated using the Analysis Trajectory tool of DS. Electrostatic potentials were calculated using the CHARMm PBEQ module implemented in DS.

## 3. Results

As mentioned above, DJ-1 is a target for many PTMs including persulfidation. Several reaction pathways may result in the persulfidation of a cysteine residue under biological conditions [39]. We, however, focused at first on the reaction of hydrogen sulfide (H_2_S) with an oxidized cysteine residue of DJ-1 and then on the reaction of a protein cysteine residue with sulfane sulfur [33] donors to access a persulfidated form of DJ-1. We carried out our reactions in a buffer containing a chelator for metal ions (DTPA), because DJ-1 has been proposed to bind copper [40,41] and zinc [42] in vitro, even if these hypotheses were recently ruled out in a cellular context [43].

### 3.1. C106 of Recombinant Human DJ-1 Is the Most Thiophilic Cysteine

To obtain persulfidated DJ-1, we first envisioned the method proposed by Pan et al. [44] based on the formation of a reactive S-S bond and its subsequent reduction by sodium hydrosulfide. Because DJ-1 possesses three cysteines, we turned our attention to 2,2’-dithiopyridine (DTP) to generate the mixed reactive disulfide thiopyridine(TP)-C106 DJ-1. Indeed, DTP should allow the selective labeling of the low- p*K_a_* C106 as it reacts with thiols (Figure 1), even at low pH [45].

The release of thiopyridine TP upon the reaction between DJ-1 and DTP can easily be monitored by UV-visible spectroscopy at 343 nm. The kinetic data obtained at pH = 7.4 (Figure 1, blue data) can be nicely fitted with a double exponential function, yielding *k_obs1_ and k_obs2_* values of 1.667 ± 0.002 and 0.088 ± 0.004 min^−1^, respectively. This observation suggests either the reaction of two of the three Cys residues of DJ-1 with DTP or the reaction of a single Cys residue with DTP followed by reduction of the mixed reactive disulfide by a second Cys residue. To distinguish between these two processes, we performed a similar experiment with the mutant C106S DJ-1. Its reactivity with DTP yields a single *k_obs_* (0.098 ± 0.006 min^−1^) similar to the slower *k_obs2_* previously measured with the wt protein, ruling out the second hypothesis. Thus, two cysteine residues of DJ-1 react with DTP, and C106 is the most thiophilic one. As expected, the selectivity for C106 is somewhat increased at pH = 6.0 for which the reaction of C46/C53 is significantly slower (*k_ob_*_s2_ = 0.034 ± 0.012 min^−1^), while the reaction of C106 is marginally faster (*k_obs1_* = 1.976 ± 0.238 min^−1^).

Next, to obtain persulfidated DJ-1, we removed excess DTP and incubated the mixed reactive disulfide TP- DJ-1 with a 20-fold excess of hydrosulfide at 25 °C for 30 min. Unfortunately, we did not observe the expected release of TP from the mixed disulfide upon sodium hydrosulfide (NaSH) treatment, indicating that the generation of a persulfidated form of DJ-1 requires another approach.

### 3.2. Recombinant Human DJ-1 Is Inactivated by Sulfane Sulfur Donors

We next turned our attention toward the reaction of DJ-1 with sulfane sulfur sources. To detect a potential modification of the protein properties, we took advantage of the esterase activity of DJ-1, which releases the chromophoric *p*-nitrophenoxide (λ = 405 nm) upon incubation with *p*-nitrophenyl acetate (*p*NPA) [18]. We started our investigation with the garlic-derived diallyltrisulfide (DATS), a natural source of reactive sulfur species (RSS) [46]. No loss of esterase activity was observed upon incubation of an excess (5–10 equiv.) of DATS with DJ-1, even after several hours. However, in the presence of an additional 1 mM of glutathione (GSH), the enzyme was dose- and time-dependently inhibited (Figure 2A), clearly advocating for the reaction between a cysteine residue of DJ-1 and (an) intermediate species formed by the reaction between GSH and DATS [47].

Among those [47,48], we first ruled out the implication of H_2_S, because it is unlikely on a mechanistic ground and because NaSH alone did not inhibit DJ-1. Interestingly, persulfides (R-SSH), generated either from the synthetic precursor P* developed in our group [49] or enzymatically produced by cystathionine γ-lyase (CSE) [50], led to a clear inhibition of DJ-1 (Figure 2B). However, glutathione persulfide, generated from GSH and phenylthiosulfonate [51], failed to inhibit DJ-1, suggesting that the steric hindrance of GSSH prevents its reactivity with a buried cysteine. Finally, the two polysulfides sodium salts di- and tetra-sulfide proved to be the most efficient donors to inhibit DJ-1. For instance, a stoichiometric amount of sulfane sulfur with respect to DJ-1 (1 equiv. of Na_2_S_2_, or 0.33 equiv. of Na_2_S_4_) leads to ~80% loss of activity after 30 min and does not further inhibit DJ-1 after 60 min, advocating for a fast reaction rate between polysulfide sodium salts and the C106 of DJ-1.

To gain further insight into DJ-1 inhibition provoked by sodium disulfide, we next carried out kinetic investigations using the approaches proposed by Mangel et al. [52] to study fast-acting proteinase inhibitors. First, we recorded the release of *p*-nitrophenoxide over time, in the presence of increasing amounts of sodium disulfide (Appendix A). As expected, a decreased plateau level was observed when the concentration of Na_2_S_2_ increased. Interestingly, for 2 to 20 µM concentrations, the rate of hydrolysis remained constant after 200 s, and approximatively three times the rate was recorded with the mutant C106S in absence of sodium disulfide, suggesting the existence of a residual activity for DJ-1 treated with Na_2_S_2_. However, we experienced difficulty trying to reproduce these experiments and extract a bimolecular rate constant for DJ-1 reactivity with sodium disulfide using this continuous activity assay, probably because of side reactions involving sodium disulfide and the substrate *p*NPA. To overcome this problem, we pre-treated DJ-1 with four different concentrations of Na_2_S_2_ for varying periods of time before recording its esterase activity. Reproducible data were thus obtained, which are presented in Figure 3A. The slope of the straight lines obtained for each Na_2_S_2_ concentration gave pseudo first-order rate constants *k_obs_* that, once plotted against the various Na_2_S_2_ concentrations, gave a bimolecular rate constant *k_inac_* of (1.69 ± 0.10) × 10^3^ M^−1^.s^−1^ (Figure 3B). It is noteworthy that a *k_inac_* of 3.8 ± 0.3 M^−1^.s^−1^ was observed in a similar experiment performed with hydrogen peroxide. The later value compares well with the one previously reported by Andres-Mateos et al. (0.56 ± 0.05 M^−1^.s^−1^), obtained using a different assay [52].

Finally, we performed experiments to assess the potential impact of Na_2_S_2_ on the protective activity of DJ-1 against methylglyoxal-induced (MGO) modification of GMP. As expected for an impaired DJ-1 activity, less lactate is produced and more GMP-MGO adduct is detected when DJ-1 is treated with the polysufide (Figure 3C). We did not investigate the exact mechanism underlying this protection, as recent kinetic studies re-assessed DJ-1 as a glyoxalase rather than a deglycase [10,53].

### 3.3. DJ-1 Is Slowly Reactivated In Vitro by DTT but Not by hTrx, hTrxR or GSH

Usually, modification of a protein by sulfane sulfur donors is reversible, and the reversal is often catalyzed by the thioredoxin and glutathione systems [54,55]. Accordingly, we monitored the reactivation of inhibited DJ-1 by various reducing systems. Incubation with DTT restores the activity of the inhibited protein, albeit with a slow *k*_reac_ of 0.075 ± 0.005 M^−1^.s^−1^ (Figure 4A). However, glutathione, the mammalian system NADPH/hTrx/hTrxR or the stoichiometric system hTrx/DTT were unable to reactivate the esterase activity of DJ-1 in our hands, pointing to the need for a specific reducing system to reactivate DJ-1. Interestingly, pre-incubation of the inhibited enzyme with H_2_O_2_ before addition of DTT or of the hTrx system did not affect the reactivation. Additionally, the oxidation of DJ-1 with H_2_O_2_, in the presence or absence of equimolar concentration of H_2_S, was irreversible regardless of the reducing agent (Figure 4A).

### 3.4. The Modification Induced by Sodium Disulfide Increases the Thermal Stability of DJ-1

Structural modifications induced on human DJ-1 by hydrogen peroxide are accompanied by a change in its melting temperature (Tm) [56], as determined by differential scanning fluorimetry (DSF), which gives information on protein stability. Accordingly, inhibited DJ-1 by sulfane sulfur donors should also exhibit a variation in its Tm as compared to native DJ-1. We thus determined the Tm of both proteins in PBS using differential scanning fluorimetry (Thermofluor). Native DJ-1 has a Tm of 60.0 °C (Figure 4B), which is identical to the value previously reported for DJ-1 under similar experimental conditions [53]. The incubation of DJ-1 (50 µM) with sodium disulfide (65 µM) shifts this Tm to 65.5 °C, advocating that the modification of DJ-1 by sodium disulfide stabilizes the protein. Interestingly, we were, under these conditions, unable to record a clean melting curve for the oxidized form of DJ-1. However, in the presence of DTT, we were able to reproduce the Tm of 75 °C previously reported [56]. The reducing agent is thus critical, and it also shifts the Tm of the persulfidated form to 70 °C but leads to a broader melting curve. Both PTMs therefore thermally stabilize DJ-1, and sulfinylation is slightly more stabilizing.

### 3.5. DJ-1 Is Persulfidated In Vitro by Polysulfides at C106

To clearly identify the modification responsible for the drop of enzymatic activity upon DJ-1 treatment with sulfane sulfur donors, we used the selective “tag-switch” method developed by Filipovic [30]. Briefly, persulfides are blocked as an activated mixed-disulfide by reaction with 4-chloro-7-nitrobenzofurazan (NBDCl), and the disulfide bond is reduced by a fluorescent dimedone derivative, leading to a fluorescent protein conveniently detected by in gel fluorescence. In the absence of Na_2_S_2_, a weak signal was detected by florescence (Figure 5A). However, incubation of DJ-1 with sodium disulfide led to a strong fluorescent spot on the gel. Interestingly, weak fluorescence was detected when the mutant C106S was incubated with or without Na_2_S_2_, indicating that DJ-1 is almost exclusively persulfidated at C106. This selectivity for C106 is confirmed by our initial data using a different assay (Appendix A).

### 3.6. Recombinant DJ-1 Is Endogenously Persulfidated in E. coli

Finally, we investigated whether recombinant DJ-1 is endogenously persulfidated when overexpressed in *E. coli* and if its persulfidation level could be influenced by sulfur compounds. When DJ-1 is overexpressed under standard conditions, persulfidation is clearly detected by in gel fluorescence assay [30] (Figure 5C). The addition of 25 mM sodium thiosulfate to the culture medium after induction not only increases DJ-1 but also the global proteins’ persulfidation levels (expressed as the Cy5/CBB signal ratio, which we found more accurate in our experiments than the Cy5/488 signal ratio proposed previously [30]), in contrast to the addition of 1 mM cysteine trisulfide, which had the opposite effect. It must be noted that endogenous persulfidation was also unexpectedly detected when overexpressing the mutant C106S in *E. coli* (Appendix A).

### 3.7. Persulfidation Affects the Sub-Unit Interface (but Sulfinylation Does Not)

In the absence of X-ray data, we decided to use molecular dynamics (MD) simulations to evaluate the impact of C106 persulfidation on the structure and the stability of the protein. Root mean square deviation (RMSD) plots (Figure 6A) show a good convergence of the trajectories after 25 ns and up to 70 ns, indicating a good stability for the two systems. The average Cα-RMSD per residue, which gives indications on the dynamics of individual amino acids of the dimer, is plotted in Figure 6B.

The difference between the plot of the persulfidated and the wt proteins highlight several differences between the two trajectories. The most prominent deviations come from a partial loss of secondary structure on the loop connecting the β7 and α5 regions and containing the C106 (residues #106–109) and in the α6 region (residues #128–138) (Appendix A) in the persulfidated DJ-1. The addition of a sulfur atom to C106 just slightly impacts the main connections between this residue and its surrounding amino acid residues (G75, S155 and R156), with very similar distances recorded between these residues in both the wt and the persulfidated form (Appendix A). As previously proposed, in the B sub-unit of the wt, the deprotonated sulfur atom of C106 is stabilized [17] by H-bond interaction with protonated E18 and G75 (distance = 2.44 Å and 2.35 Å, respectively). In sub-unit A, the sulfur atom does not interact with E18, a feature already reported by others [57]. Similarly, a weak interaction between E18 and the internal sulfur atom of the persulfide is noticed in the B sub-unit of the persulfidated form (distance = 3.96 Å). However, the outer sulfur atom of the persulfide remains mostly unstabilized and only interacts in a few conformations with the NH moiety of G75 (average distance of 3.78 Å). In addition, the latter is replaced in a few rare cases by an H-bond contact between G108 and the carbonyl group of C106 (Appendix A). The poor interaction network of the terminal sulfur may explain its higher fluctuation, as shown by its higher root mean square fluctuation (RMSF) value compared to the one calculated for the inner sulfur atom of the persulfidated DJ-1 or the sulfur atom from C106 in the wt protein (1.53 vs. 0.57 and 0.80 Å, respectively) [58]. The same trend is followed for solvent accessibility, the additional sulfur atom being the most accessible one (SAS of 33.4, 7.5 and 11.8 Å^2^ for the additional S atom, the S atom of the persulfide and the wt forms of DJ-1, respectively). Additionally, the persulfidation of C106 impacts the interface between the two sub-units. Thus, crucial interactions implicated in the structural stabilization of the wt dimer [59,60] are either lost (R27A:R48B, G159A:L185B, Appendix A) or weakened (R28A:E15B, E18A:R28B and D49A:R27B, Appendix A) in persulfidated DJ-1 when compared to the wt. Moreover, a new interfacial H-bond contact appears between the guanidine group of R28 from the B sub-unit and the hydroxyl moiety of S47 from the A sub-unit (average distance of 3.15 Å vs. 7.40 Å on the last 45 ns in the persulfidated vs. wt, Appendix A). Overall, a significant decrease in the average interfacial interaction energy calculated on the last 20 ns is observed between persulfidated DJ-1 (−288.14 ± 16.87 kcal.mol^−1^) and the wt protein (−319.67 ± 12.89 kcal.mol^−1^). Finally, noticeable changes both in the hydrophobicity and electrostatic potentials are observed between the wt and the modified protein, in particular around the α6 helix that is the region showing the highest RMSD between the wt and the modified protein (Appendix A). Interestingly, C106 becomes more accessible after persulfidation, thus suggesting functions related to this exposure, e.g., intervention in transpersulfidation reactions or reversibility of the modification thanks to a dedicated de-persulfidation system, while the SASA of the protein remain unchanged (14,706 ± 175 vs. 14,979 ± 170 Å^2^).

Next, to compare persulfidation and sulfinylation, we generated the sulfinylated form of DJ-1 and performed MD simulations using the same procedure. A comparison between the final conformations of both oxidized proteins is presented in Appendix A. The structures of the persulfidated and sulfinylated DJ-1 are globally similar, but the latter does not show the partial loss of secondary structure observed in the α6 and α5-β7 regions of the former. Importantly, the active sites differ significantly, with strong interactions detected in the sulfinylated protein between the backbone of His126 and either the SO_2_ moiety (sub-unit A) or the carbonyl group of C106 (sub-unit B) (Appendix A). An additional H-bond contact between the sulfinic group and E18 additionally locks the conformation of C106 in sub-unit A, while G157 and H126 are in H-bond radius with the sulfinyl group in the B sub-unit. In addition, S155 strongly interacts with C106 backbone in the B sub-unit. Thus, the interactions observed in the persulfidated and wt forms between G75 and C106 are lost in the sulfinylated DJ-1. These results are in very good agreement with a previous study on oxidized and overoxidized DJ-1 [57]. This new set of interactions results in a less mobile cysteine residue (the average RMSF of C106 is 0.98 Å for persulfidated DJ-1 and 0.53 Å for the sulfenylated form). Despite these significant changes in the active site, none of the interfacial changes observed upon persulfidation are detected in the sulfinylated form: the interaction R27A:R48B is even stronger than in the wt, while other important interactions, G159:L185, D49:R27, R28:E15, S47:R28, remain intact (Appendix A). Additionally, various changes are observed at the protein surface (Appendix A).

## 4. Discussion

As the only thiol-containing residue of proteins, cysteine is a crucial player in redox sensing and signaling. For instance, the redox messenger hydrogen peroxide [61] and the gaseous transmitter nitric oxide [62] have been long known to signal at least partially through the modification of cysteine residues. More recently, hydrogen sulfide, a gaseous transmitter endogenously produced via enzymatic activity, has also been reported to signal through a post-translational modification of cysteine, first named S-sulfhydration [63] but more rigorously re-denominated persulfidation [64]. This PTM, which converts a cysteine residue CysSH into the corresponding persulfide CysSSH, may result from the reaction between H_2_S and cysteine sulfenic acid (CysS-OH), the H_2_O_2_-oxidized form of cysteine [65]. In contrast, CysSSH may also be formed by the reaction between CysSH and bound sulfane sulfur species (BSS) [66]. BSS, which encompass sulfur-derivatives with a sulfur formal oxidation state of −I and 0 [66], are notably produced in cells by hydrogen sulfide [67] or 3-mercaptopyruvic acid [68] metabolism. Although their exact speciation is challenging to establish, their global concentrations (from high nM to 50–100 µM, depending on the studies) [49,66] are orders of magnitude higher than those detected for hydrogen sulfide or H_2_O_2_ in various biological media under physiological conditions [66].

The outcome of persulfidation of cysteine residues on the function of targeted proteins is quite diverse [69,70] but sometimes conflicting. For instance, the first protein reported to be persulfidated (GAPDH) has been proposed to be activated or inhibited by this PTM [63,71]. Persulfidation may also induce intra-cellular relocation of targeted proteins, as observed with persulfidated GAPDH that is re-distributed into the nucleus, enabling it to participate in H_2_S-mediated activation of autophagy [72]. Finally, this modification also regulates protein–protein interactions, as detected for the Keap1/Nrf2 system during the activation of the antioxidant response [73]. In addition to its role in intra-cellular signaling cascades, persulfidation has also recently been proposed as a protective mechanism against irreversible cysteine overoxidation during oxidative stress. Hydrogen sulfide may quench reactive sulfenic acid intermediates, thus preventing their further oxidation into sulfinic/sulfonic acids and allowing the resulting persulfide (or persuf(e,i,o)nic species) to be reduced back to the thiol status by glutathione or the thioredoxin system [54,55].

DJ-1 has been known to be sulfinylated for years, with both C106 [22,23] and C46 [26,74] being target cysteines. The sulfinylation of C106 acts as a redox switch for DJ-1 activity, while the role of the modification of C46 is still obscure but likely of physiological significance since C46-SO_2_H has been described to be a substrate for sulfiredoxin Srx. More recently, DJ-1 has also been shown to experience persulfidation in mammalian cell lines [30], but neither the involved cysteine(s) nor the implications of this PTM on DJ-1 structure or activity have been analyzed. Accordingly, we aimed to elucidate the exact nature and consequences of the persulfidation process on DJ-1. At first, we expressed human DJ-1 in *E. coli* and confirmed that it is partially persulfidated. The persulfidation level of DJ-1 increased in the presence of thiosulfate, a sulfur source for *E. coli*, but reduced in the presence of cysteine trisulfide, which has recently been proposed to be metabolized by *E. coli* into cysteine hydropersulfide [36]. Therefore, we expected to detect higher persulfidation yield of DJ-1 with cysteine trisulfide. However, a recent report confirms that cysteine trisulfides acts as an oxidative species leading principally to the oxidation of cysteine residues into mixed di- or trisulfides [75], which would account for the observed decrease in persulfidation levels when using this sulfane source. Next, the persulfidated form of DJ-1 was obtained by reacting purified wt DJ-1 with various sulfane sulfur donors, since the reduction of the activated disulfide bond formed from C106 and DTP by sodium hydrosulfide [43] had failed. Donors included cysteine hydropersulfide (enzymatically produced from cystine by CSE) or the polysulfide Na_2_S_2_ and Na_2_S_4_, used at physiologically relevant concentrations. Interestingly, glutathione hydropersulfide (formed in situ from glutathione and a chemical sulfur donor) [50] did not react with wt DJ-1, indicating that the size and/or charge of the sulfane sulfur donor govern the access to the reactive cysteine(s). Under these conditions, DJ-1 is selectively persulfidated at C106, as confirmed by the weak reactivity of the C106S mutant with sodium disulfide. This agrees well with our observations indicating that C106 is the most thiophilic Cys residue of DJ-1 when reacted with 2,2’-dithiopyridine (DTP) and with a previous study showing that glutathionylating agents mainly modify C106 [76]. This, however, contrasts with our observations suggesting that the mutant protein is also endogenously persulfidated in *E. coli*, even if the persulfidation of endogenous YajL, a member from the DJ-1 superfamily from *E. coli* [6], may account for this observation.

The post-translational oxidation of C106 by polysulfides inhibits DJ-1 C106-based activities (esterase or deglycase/glyoxalase activities). However, contrary to the sulfinylation that irreversibly inhibits the enzyme, the persulfidation is slowly reversible. Furthermore, in contrast to persulfidated PTP1B [77], HAS [54] or BSA [55] which are reactivated by the Trx and/or Grx systems, the persulfide of DJ-1 is solely reduced by DTT. This hints at a poor accessibility or electrophilicity of the inner sulfenyl sulfur of persulfidated C106. This would not only explain the lack of reactivity of H_2_S with the reactive mixed-disulfide form between C106 and DTP but also the absence of reactivity of the sulfenic acid of DJ-1 toward hydrogen sulfide (see below). This hypothesis, in agreement with the trans-nitrosylation (rather than the formation of a disulfide bond) observed from DJ-1 to the phosphatase PTEN [28], is also supported by MD simulation showing a larger solvent accessibility of the terminal sulfur atom.

To gain insights into the kinetics of the sulfur transfer during the persulfidation of DJ-1 by small molecules, we focused on the reaction between DJ-1 and sodium disulfide, since the generation of the reactive persulfide from the systems CSE/cystine, P* or DATS/GSH are slow and limited by the rate of formation of the donor. We thus determined a rate constant of (1.69 ± 0.10) × 10^3^ M^−1^.s^−1^, which is, to our knowledge, the first reported for the reaction between a cysteine residue and a polysulfide. It is orders of magnitude higher than the bimolecular rate constant determined for the reaction of DJ-1 with hydrogen peroxide. Indeed, despite the low p*K_a_* of C106, its oxidation by hydrogen peroxide is slow (k = 0.56 M^−1^.s^−1^ [15] or 3.8 M^−1^.s^−1^ in this study, at pH 7.4) but in the range of those reported for the low molecular weight cellular thiols glutathione, cysteine or hydrogen sulfide (k = 0.9, 2.9 and 0.73–15 M^−1^.s^−1^ at pH 7.4, respectively) [78,79,80] or for proteins such as human serum albumin (HSA) (k = 2.7 M^−1^.s^−1^) [81]. Additionally, DJ-1 does not stabilize the sulfenic form that quickly oxidizes to the sulfinic form [23].

Therefore, contrary to other proteins in which the sulfenic acid is stabilized and may be quenched by hydrogen sulfide [65,82], the persulfidation of DJ-1 via the formation of its sulfenic acid (Figure 2, red arrow) is, in our opinion, less likely than its reaction with polysulfides (Figure 2, green arrow). This view is supported by our experiment showing that DJ-1 is fully and irreversibly inhibited in the presence of equimolar concentrations of H_2_O_2_ and H_2_S.

Finally, because C106 is the target of both sulfinylation and persulfidation, we next used additional techniques to investigate the differences induced by these two PTMs. The persulfidation of DJ-1 slightly stabilized the protein thermally compared to the wt form, as indicated by the 5 °C variation of their respective Tm. Unfortunately, we were unable to obtain a clean melting curve with the sulfinylated form of DJ-1 and to directly compare the inherent stabilization afforded by each PTM, most likely because we had to work in the absence of DTT, which reduces intra-disulfide bridges formed upon the oxidation of DJ-1 [56,74]. Indeed, the determination of the Tm of various forms of DJ-1 is highly sensitive to the experimental conditions. For instance, totally different melting curves have been reported for the overoxidized form of DJ-1 [57,83]). Our result would nevertheless suggest that the intrinsic stability of DJ-1 oxidized by Na_2_S_2_ or H_2_O_2_ differs and that their tertiary structures are dissimilar, which is supported by the lack of detection of persulfidated DJ-1 by the antibody directed against DJ-1 harboring a sulfinate. However, our MD studies indicate that both PTMs lead to tridimensional structures similar to the wt, a result already reported for sulfinylated and sulfonylated DJ-1 [57,84]. Nevertheless, at the local level, persulfidation results in a partial loss of the secondary structure and a decrease in the interfacial interaction energy similar to those observed in the pathological mutants such as A104T [59] but absent in the sulfinylated form.

## 5. Conclusions

In conclusion, in this work, we confirmed that DJ-1 is persulfidated not only in mammalian cells but also in *E. coli*. This PTM implicates cysteine C106, and on the basis of kinetic studies, we propose that this oxidation takes place by the reaction of C106 with sulfane sulfur donors rather than by the reaction of its sulfenic form with hydrogen sulfide. Like sulfinylation, persulfidation inhibits two C106-based activities, but the activity of the latter may be recovered in the presence of a reductant, albeit slowly. Additionally, various data suggest a structural difference between these two PTMs, which could both play a dedicated role in DJ-1 signaling or protective pathways or protein–protein interactions.

## Data Availability

Raw gel images are available.

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
