# Peer review of "Persulfidation of DJ-1: Mechanism and Consequences"

_biomolecules, 2022, doi:10.3390/biom13010027_

Round 1

Reviewer 1 Report

The authors report a predominantly in vitro study of the persulfidation of DJ-1, a biomedically important protein whose function(s) have remained controversial.  The functional importance of Cys106 is widely corroborated, and there has been sustained interest in understanding how posttranslational modification (PTM) of that residue modulate DJ-1 function.  Although oxidation to the sulfinate/sulfonate has been the most studied, there are reliable reports that DJ-1 is persulfidated, although the details of this PTM are not know.  The authors address this gap in knowledge by performing a careful study of DJ-1 persulfidation, determining that it occurs predominantly at Cys106, is orders of magnitude faster than oxidation, and can be observed in cells.  Overall, this is a well-conducted study that adds valuable quantitative data to the research literature.  The manuscript is well-written and illustrated, and most of my comments are minor.   I detail these below.

Major Comments:

1.      The authors observe that oxidation of Cys106 leads to a messy thermal denaturation curve in Fig. 4B.  As the authors note, they are using a protocol that has been reported to generate Cys106-SO2-, which has typically been found to stabilize DJ-1.  However, oxidation to Cys106-SO3- has been reported to destabilize the protein, as discussed in lines 585-588.   It seems likely to this reviewer that the authors have generated a mixture of SO2-/SO3- in this sample. The authors should characterize the oxidation state of their protein using mass spectrometry and, if necessary, adjust their oxidation conditions to produce predominantly the species of interest. 

2.     In various places in the manuscript, results are reported as “data not shown” (e.g. lines 326, 396, etc).  In the age of supplemental data (which the authors have used to good effect elsewhere), this seems strange.  I strongly recommend that any results that are being reported also be shown in Supplemental.

Minor comments:

3.     Line 88, Cary 300 is misspelled.

4.     Line 94, protein expression is cited as previously described, but no reference is given-it should be provided. 

5.     Line 164, NBD is used to detect persulfidation in cell extract, but I also thought that it detects sulfenylation.  If this is correct, it would be helpful for the authors to clarify specificity of the probe for the persulfidated modification.

6.     Line 186; the primary literature reference for PDB 3SF8 should be provided.

7.     Line 301, glutathione is misspelled.

8.     Lines 359-360.  Why did the authors not make this comparison directly to the experimentally determined structure of Cys106-SO2- DJ-1 (PDB 1SOA), rather than using a generated model?

9.     Line 473, I believe “sulfinylated” is meant here.

Author Response

REVIEWER 1

  1. The authors observe that oxidation of Cys106 leads to a messy thermal denaturation curve in Fig. 4B. As the authors note, they are using a protocol that has been reported to generate Cys106-SO2-, which has typically been found to stabilize DJ-1.  However, oxidation to Cys106-SO3- has been reported to destabilize the protein, as discussed in lines 585-588.   It seems likely to this reviewer that the authors have generated a mixture of SO2-/SO3- in this sample. The authors should characterize the oxidation state of their protein using mass spectrometry and, if necessary, adjust their oxidation conditions to produce predominantly the species of interest.

We have now run the experiment in the presence of DTT, which give results similar to those previously described. Inter-molecular disulfide bridges are most probably responsible for the messy curve obtained without DTT.

  1. In various places in the manuscript, results are reported as “data not shown” (e.g. lines 326, 396, etc). In the age of supplemental data (which the authors have used to good effect elsewhere), this seems strange.  I strongly recommend that any results that are being reported also be shown in Supplemental.

We have now added a new figure for the GMP protection studies.

Minor comments:

  1. Line 88, Cary 300 is misspelled.

This has been corrected

  1. Line 94, protein expression is cited as previously described, but no reference is given-it should be provided.

This has been corrected

  1. Line 164, NBD is used to detect persulfidation in cell extract, but I also thought that it detects sulfenylation. If this is correct, it would be helpful for the authors to clarify specificity of the probe for the persulfidated modification.

NBD-Cl indeed reacts with sulfenic acids to yield a sulfone. Persulfides yields disulfides, which are then reduced by the fluorescent-dimedone like probe, leading to Cy5 fluorescence. Therefore, even if sulfenic acids will react with NBD-Cl, they will not lead to Cy-5 fluorescence.

  1. Line 186; the primary literature reference for PDB 3SF8 should be provided.

This has been included

  1. Line 301, glutathione is misspelled.

This has been corrected

  1. Lines 359-360. Why did the authors not make this comparison directly to the experimentally determined structure of Cys106-SO2- DJ-1 (PDB 1SOA), rather than using a generated model?

We decided to be fully consistent. Moreover the protocol we used was also used by the authors of ref 57, allowing us to validate our model.  

  1. Line 473, I believe “sulfinylated” is meant here

It should have been sulfinic.  Thanks.

Reviewer 2 Report

It is a very interesting and valuable work written in a well-readable format. 

My only major criticism is the complete lack of statistical analysis. It is clear, that presented dot-plots and curves support the explanation/conclusion of the authors, however, some results, especially which are presented in bar charts are difficult to evaluate. I would like to highlight Fig 5/right, where results are presented as mean +/- SE, and interpreted as S2O3 increased persulfidation level. In the absence of statistical analysis or showing individual values beneath bars, the comparison of groups is questionable. I would highly recommend to complete the manuscript with appropriate statistical analysis.

Figures à

·       Figure styles should be unified. Some of them are presented as Figure X/left, /right, others /A, /B, Figure 5 contains both divisions.

·       In Figure 1, authors compare wt and mutant DJ-1 proteins. In the following figures, the type of the investigated DJ-1 is not defined. Figure legends should be completed with the corresponding adjectives.

·       In Figure 2 (left), dot plot should be completed with connecting lines in order of better visualisation and interpretation, similar as in Figure 1, 3, 4, etc.

·       In Figure 2 (left), symbol legends should be showed in the figure as well, similar as in other figures.

Minor comments à

·       line 12: typo (Cys116 vs Cys106)

·       line 30: pathological examples should be completed with other diseases, associated with oxidative conditions (e.g. intestinal diseases with chronic inflammation, diabetes, etc).

·       overall: nomenclature of cysteine residues should be unified (Cys106 vs C106)

·       line 59: Cter abbreviation should be defined in the previous sentence

·       line 144: “80% drop..” should be discussed in results section, and/or should be supported by literary or own data.

·       line 148: DSF abbreviation should be defined, and the aim of the method should be mentioned in this section (e.g. to detect protein stability/denaturation, differential scanning fluorimetry was performed)

Results section is well-readable and informative completed with explanations and comments, which could really help the understanding of the extracted information and findings. However, the first block of results (line 212-219) describing the experimental conditions should be placed to the Discussion section.

Questions à

·       Authors note at multiple times, that DJ-1 protein has various enzymatic activities, including esterase, deglycase, glyoxalase activites. In addition, many other biological functions of DJ-1 were described (chaperon, antioxidant, regulation of transcription, regulation of signal transduction pathways, etc.), and these functions are highly dependent from the oxidative status of Cys106. In this study, the biological activity of DJ-1 was only determined based on its esterase activity. In one hand, in line 275 results title should be specified: the esterase activity of DJ-1 was inactivated by sulfane sulfur donors. Did the authors investigate the other biological activity of wt and mutant DJ-1 proteins? The antioxidant, glyoxalase and deglycase activity of recombinant proteins can be investigated in very simple in vitro experiments (cleavage of H2O2, methylglyoxal or glycated proteins). In my opinion this study could be strengthen further by completing it with similar functional experiments.

·       Kitamura et al. identified small molecules which can selectively bind to the reduced Cys106, protecting DJ-1 from overoxidation and thereby preserving its biological activity. These compounds (Comp23, UCP0054277, UCP0054278) were efficient in vitro, cellular and in vivo experiments in neuronal and intestinal cells in numerous studies. Usage of this compounds could support the findings of the authors about the specific persulfidation of Cys106 compared to other Cys residues.

·       As the authors mention, DJ-1 forms homodimers and dimerization of the proteins depends on the state of Cys106. Did the authors investigate the dimerization affinity of wt and mutant, or sulfinated DJ-1 from their experiments? Did the authors detect DJ-1 dimers after gel electrophoresis in non-reducing conditions? And in case of DFS?

Author Response

It is a very interesting and valuable work written in a well-readable format. 

My only major criticism is the complete lack of statistical analysis. It is clear, that presented dot-plots and curves support the explanation/conclusion of the authors, however, some results, especially which are presented in bar charts are difficult to evaluate. I would like to highlight Fig 5/right, where results are presented as mean +/- SE, and interpreted as S2O3 increased persulfidation level. In the absence of statistical analysis or showing individual values beneath bars, the comparison of groups is questionable. I would highly recommend to complete the manuscript with appropriate statistical analysis.

Statistical analysis is now in Fig. 2,3 & 5.

Figures à

All changes have been done.

Minor comments à

  • line 12: typo (Cys116 vs Cys106)

This has been corrected

  • line 30: pathological examples should be completed with other diseases, associated with oxidative conditions (e.g. intestinal diseases with chronic inflammation, diabetes, etc).

This has been corrected

  • overall: nomenclature of cysteine residues should be unified (Cys106 vs C106)

 This has been corrected

  • line 59: Cter abbreviation should be defined in the previous sentence

This has been corrected

  • line 144: “80% drop..” should be discussed in results section, and/or should be supported by literary or own data.

 This is supported by Figure 2B. One sentence was added.

  • line 148: DSF abbreviation should be defined, and the aim of the method should be mentioned in this section (e.g. to detect protein stability/denaturation, differential scanning fluorimetry was performed)

This has been corrected . One sentence was added in the results part.

Results section is well-readable and informative completed with explanations and comments, which could really help the understanding of the extracted information and findings. However, the first block of results (line 212-219) describing the experimental conditions should be placed to the Discussion section.

We would like to keep this part as it was. This would require a large change in the discussion to introduce it.

Questions à

  • Authors note at multiple times, that DJ-1 protein has various enzymatic activities, including esterase, deglycase, glyoxalase activites. In addition, many other biological functions of DJ-1 were described (chaperon, antioxidant, regulation of transcription, regulation of signal transduction pathways, etc.), and these functions are highly dependent from the oxidative status of Cys106. In this study, the biological activity of DJ-1 was only determined based on its esterase activity. In one hand, in line 275 results title should be specified: the esterase activity of DJ-1 was inactivated by sulfane sulfur donors. Did the authors investigate the other biological activity of wt and mutant DJ-1 proteins? The antioxidant, glyoxalase and deglycase activity of recombinant proteins can be investigated in very simple in vitro experiments (cleavage of H2O2, methylglyoxal or glycated proteins). In my opinion this study could be strengthen further by completing it with similar functional experiments.

      This is currently under investigation in the lab, especially for the part dedicated to non-enzymatic DJ-1 role.

  • Kitamura et al. identified small molecules which can selectively bind to the reduced Cys106, protecting DJ-1 from overoxidation and thereby preserving its biological activity. These compounds (Comp23, UCP0054277, UCP0054278) were efficient in vitro, cellular and in vivo experiments in neuronal and intestinal cells in numerous studies. Usage of this compounds could support the findings of the authors about the specific persulfidation of Cys106 compared to other Cys residues.

We have been using isatin so far in this context.

  • As the authors mention, DJ-1 forms homodimers and dimerization of the proteins depends on the state of Cys106. Did the authors investigate the dimerization affinity of wt and mutant, or sulfinated DJ-1 from their experiments? Did the authors detect DJ-1 dimers after gel electrophoresis in non-reducing conditions? And in case of DFS?

Yes, you can see traces of dimers on the gels submitted as raw data. However, we are not sure whether or not they are relevant to physiological conditions were the concentrations of DJ-1 are supposedly lower that those used in our experimental conditions.

Reviewer 3 Report

In the paper ‘Persulfidation of DJ-1: Mechanism and Consequences’, Galardon and colleagues analyzed the persulfidation reaction of recombinant DJ-1 induced with various sulfane sulfur and analyzed the effect of persulfidation on the structure and esterase activity of the protein in vitro. Furthermore, the authors suggest that the Cystein residue C106 is the main target of persulfidation. This would be an interesting finding as C106 is a highly conserved residue and in vivo subject to several oxidation steps that facilitate ROS sensing and translocation of DJ-1 to the mitochondria. This functionally essential oxidation of C106 can however result in sulfonic DJ-1, which is non-functional. Persulfidation is suggested to protect proteins from this hyperoxidation and loss of function. Although the paper is well written and nicely structured, it has major weaknesses in study design and interpretation of the results. Especially the lack of any statistical analysis throughout the paper is bad scientific practice. Furthermore, studying posttranslational modifications of an mammalian protein by overexpressing it in prokaryotes is prone to produce experimental artifacts. Hence, the conclusions drawn for biological relevance in this study are at best questionable.

Major remarks:

11)     Fig. 2: The authors try various sulfane sulfur donors and highlight DATS that only leads to a decrease of esterase activity of DJ-1 when co-incubated with GSH (formation of a new donor). However, the physiologically most relevant donor H2S shows no effect. It is suggested that persulfidation in vivo with H2S requires prior oxidation of the Cys residue, which is usually caused by endogenous H2O2. IN the present study it is not clear which oxidative state the Cys106 residue of the recombinant protein has. The authors should have co-treated the recombinant DJ-1 with various concentrations of H2O2. The search for more potent donors is chemically interesting, however does only make sense if the authors at least discuss the biological relevance of the chosen donors in a cellular context.

22)     Fig 2: Statistical analysis is missing.

33)     Line 334 – line 338: There are no data shown for this. This paragraph refers to no figure.

44)     Fig 5: Overexpressing tagged DJ-1 in E. coli to study persulfidation falls way back behind the state-of-the-art. DJ-1 is a ubiquitously expressed and highly abundant protein in mammalian cells. There is no need to overexpress it in E. coli to study it’s persulfidation. In fact, Filipovic and colleagues who invented the method used here, have already shown persulfidation of DJ-1 in mammalian cells. The authors should ideally use human iPSC-derived neurons to study wt DJ-1 and engineered C106S iPSC-derived neurons to study the mutant protein. If the experimental expertise for this is not given, they should use at least mammalian cell lines.

55)     Fig 5A: The result is quite puzzling. The authors did not use a YajL knockout strain (the E. coli homologue of DJ-1 with a conserved Cys residue similar to the Cys106) to obtain clean results. YajL has a size comparable to DJ-1 and would give a signal, too. This might explain why they detect a signal without addition of Na2S2. But that the image presented does even show the disappearance of the signal in the C106S strain after Na2S2 addition, does not make sense. In the original image 1 out of three experiments still gives a signal (lane 6), while 2 don’t. The one that gives a signal is the same extract as in lane 2 but just 50% was loaded. In lane 10 the the concentration of Na2S2 was increased by 50% and again only half the amount loaded compared to lane 2 and this again doesn’t give a signal. This looks a bit like a technical with the experiments or is there another explanation for this. The authors should perform this experiment in multiple replications and quantify the bands and provide a summary with statistical analysis. As presented now, the authors have performed 2 experiments (100 and 150 µM Na2S2) once and draw conclusions from n=1.

66)     Fig 5B: Statistical analysis is missing.

77)     Fig 5B: It is not acceptable to state that there is also a signal detected for the mutant and not show the data. The main point the authors want to make is that persulfidation of DJ-1 happens predominantly at Cys106 and is therefore od biological relevance. The reader must be able to assess the results themselves and need therefore the data. The authors have to also statistically analyze the results of wt vs. C106S to claim that there is a difference in persulfidation. But as mentioned above, all these experiments should be performed in a physiologically more relevant model (at least mammalian cell line) with endogenous DJ-1.

Minor comment:

Line 349: A typo, it should be “in the presence or absence of”

Author Response

Major remarks:

11)     Fig. 2: The authors try various sulfane sulfur donors and highlight DATS that only leads to a decrease of esterase activity of DJ-1 when co-incubated with GSH (formation of a new donor). However, the physiologically most relevant donor H2S shows no effect. It is suggested that persulfidation in vivo with H2S requires prior oxidation of the Cys residue, which is usually caused by endogenous H2O2. IN the present study it is not clear which oxidative state the Cys106 residue of the recombinant protein has. The authors should have co-treated the recombinant DJ-1 with various concentrations of H2O2. The search for more potent donors is chemically interesting, however does only make sense if the authors at least discuss the biological relevance of the chosen donors in a cellular context.

We checked the oxidative state of C106 at the start of the experiments and no oxidation of the protein was detected using anti-oxDJ-1 antibody.

22)     Fig 2: Statistical analysis is missing.

This is now done In Fig 2,3 &5.

33)     Line 334 – line 338: There are no data shown for this. This paragraph refers to no figure.

This is now corrected. We have re-performed the experiments with a newly acquired much more sensitive LC-MS setup. Experimental data and results are now in the manuscript.

44)     Fig 5: Overexpressing tagged DJ-1 in E. coli to study persulfidation falls way back behind the state-of-the-art. DJ-1 is a ubiquitously expressed and highly abundant protein in mammalian cells. There is no need to overexpress it in E. coli to study it’s persulfidation. In fact, Filipovic and colleagues who invented the method used here, have already shown persulfidation of DJ-1 in mammalian cells. The authors should ideally use human iPSC-derived neurons to study wt DJ-1 and engineered C106S iPSC-derived neurons to study the mutant protein. If the experimental expertise for this is not given, they should use at least mammalian cell lines.

This is now under study in our lab and in our opinion it falls beyond the scope of this biochemical preliminary study. Our approach is to start from the biochemical characterization of the enzyme and to then move on to cellular experiments, then to in vivo models. 

55)     Fig 5A: The result is quite puzzling. The authors did not use a YajL knockout strain (the E. coli homologue of DJ-1 with a conserved Cys residue similar to the Cys106) to obtain clean results. YajL has a size comparable to DJ-1 and would give a signal, too. This might explain why they detect a signal without addition of Na2S2. But that the image presented does even show the disappearance of the signal in the C106S strain after Na2S2 addition, does not make sense. In the original image 1 out of three experiments still gives a signal (lane 6), while 2 don’t. The one that gives a signal is the same extract as in lane 2 but just 50% was loaded. In lane 10 the the concentration of Na2S2 was increased by 50% and again only half the amount loaded compared to lane 2 and this again doesn’t give a signal. This looks a bit like a technical with the experiments or is there another explanation for this. The authors should perform this experiment in multiple replications and quantify the bands and provide a summary with statistical analysis. As presented now, the authors have performed 2 experiments (100 and 150 µM Na2S2) once and draw conclusions from n=1.

We have added a comment in the manuscript about YajL. In the manuscript, we used the term “selectivity” rather than “specificity” in the manuscript, implying that persulfidation may also occur at other cysteine(s). Finally, we also added other data using another persulfide assay (Figure S’1), which also shows selectivity for C106.

66)     Fig 5B: Statistical analysis is missing.

      This is now corrected.

77)     Fig 5B: It is not acceptable to state that there is also a signal detected for the mutant and not show the data. The main point the authors want to make is that persulfidation of DJ-1 happens predominantly at Cys106 and is therefore od biological relevance. The reader must be able to assess the results themselves and need therefore the data. The authors have to also statistically analyze the results of wt vs. C106S to claim that there is a difference in persulfidation. But as mentioned above, all these experiments should be performed in a physiologically more relevant model (at least mammalian cell line) with endogenous DJ-1.

Minor comment:

Line 349: A typo, it should be “in the presence or absence of”

      This is now corrected.

Reviewer 4 Report

The study is very interesting. Perhaps you can check if Glutaredoxins can reverse the modification.

Author Response

The study is very interesting. Perhaps you can check if Glutaredoxins can reverse the modification.

Thank you. This is under consideration, in more complex sets of experiments.

Reviewer 5 Report

DJ-1 (also called PARK7) is a ubiquitously expressed protein involved in the etiology of Parkinson disease. Cys 106 of DJ-1 should play a pivotal role, and its oxidation state might determine the specific function of the enzyme. DJ-1 was recently reported to be persulfidated in mammalian cell lines, but the implications of this post-translational modification have not yet been analyzed. The authors reported that recombinant DJ-1 is reversibly persulfidated at Cys116 by reaction with various sulfane donors and subsequently enzymatic function of DJ-1 was inhibited. Strikingly, this reaction is orders of magnitude faster than Cys106 oxidation by H2O2. Persulfidated of DJ-1 at Cys116 likely play a dedicated role in DJ-1 signaling or its protective function.

Although a vast number of papers have been reported on the molecular function of DJ-1, genuine molecular function of DJ-1 is still unknown. One of the reasons for confusion about molecular function of DJ-1 is that some papers poorly have reproducibility and are unreliable. This paper is descriptive and thus has little novelity. Nevertheless, this manuscript seems to be supported by reliable biochemical analyses and is excellent. I thus strongly support the publication of this manuscript, although the following two points need to be addressed.

(1) Figure 3
In this manuscript, the authors stated the results and methods as follows:
"Finally, we verified that the reaction between DJ-1 and sodium disulfide also inhibits the protective activity of DJ-1 against glycation by methylglyoxal. As expected, while the purified enzyme prevents 70% of guanosine monophosphate (GMP) glycation vs a control experiment without enzyme, the modified-enzyme does not exhibit any protective activity." (Results)

"The protection against guanine glycation in the presence of methylglyoxal by DJ-1 was determined by High Performance Liquid Chromatography as previously reported". (Methods)

Nevertheless, the data from the above-mentioned experiments were not presented in the manuscript. Namely, the original data suggesting that 'sodium disulfide inhibits the protective activity of DJ-1 against glycation by methylglyoxal' are shown neither in the main figures nor in the supplementary figures. The experimental results should be presented in the figure.

Furthermore, when describing these results, the authors should pay attention whether the experimental results are really derived from the deglycase activity of DJ-1. Because Dr. Dairou (the final author of this manuscript) is the author of the previous paper entitled 'Guanine glycation repair by DJ-1/Park7 and its bacterial homologs’ (Science 2017), he should believe in the deglycase activity of DJ-1. However, following the reliable papers by Dr. Wilson (https://pubmed.ncbi.nlm.nih.gov/35713360/) and Dr. Utepbergenov (https://pubmed.ncbi.nlm.nih.gov/31653696/), deglycase activity is a secondary effect that is derived from glyoxalase activity against MG that is in rapid equilibrium with reversible MG adducts.
When explain the above results, the authors should be very careful whether the results are derived from deglycase activity of DJ-1 or from the secondary effects of glyoxalase activity of DJ-1.

(2) Fig. 4B

The authors showed the interesting results that sodium disulfide increases the thermal stability of DJ-1. This result is very interesting for me. At the same time, the authors reported that they were unable to observe a clean melting curve for the oxidized form of DJ-1 under their experimental DSF conditions. In addition, they could not observe stabilization of DJ-1 following incubation with hydrogen peroxide.

However, I had independently performed the thermal shift experiments of DJ-1, and confirmed that H2O2 stabilizes DJ-1 very clearly as has been reported previously (Ref 53). I think the best and the easiest way to detect the thermal stabilization of DJ-1 is using Protein Thermal Shift Assay Kit of Thermo Fisher Scientific.

Using that kit (Thermo Fisher Scientific, catalog # 4461146), H2O2-dependent stabilization of DJ-1 can be easily checked according to the manufacturer's protocol. The authors should perform such the experiment.

Author Response

 We thank the reviewer for his critical comments.

(1) Figure 3
In this manuscript, the authors stated the results and methods as follows:
"Finally, we verified that the reaction between DJ-1 and sodium disulfide also inhibits the protective activity of DJ-1 against glycation by methylglyoxal. As expected, while the purified enzyme prevents 70% of guanosine monophosphate (GMP) glycation vs a control experiment without enzyme, the modified-enzyme does not exhibit any protective activity." (Results)

"The protection against guanine glycation in the presence of methylglyoxal by DJ-1 was determined by High Performance Liquid Chromatography as previously reported". (Methods)

Nevertheless, the data from the above-mentioned experiments were not presented in the manuscript. Namely, the original data suggesting that 'sodium disulfide inhibits the protective activity of DJ-1 against glycation by methylglyoxal' are shown neither in the main figures nor in the supplementary figures. The experimental results should be presented in the figure.

We have remodeled the manuscript to include these data. Full experimental details are now given, along with experimental results (Figure 3c). We have re-run the experiments using a different setup, using a newly acquired LC-MS which is much more sensitive than the previous test.        

Furthermore, when describing these results, the authors should pay attention whether the experimental results are really derived from the deglycase activity of DJ-1. Because Dr. Dairou (the final author of this manuscript) is the author of the previous paper entitled 'Guanine glycation repair by DJ-1/Park7 and its bacterial homologs’ (Science 2017), he should believe in the deglycase activity of DJ-1. However, following the reliable papers by Dr. Wilson (https://pubmed.ncbi.nlm.nih.gov/35713360/) and Dr. Utepbergenov (https://pubmed.ncbi.nlm.nih.gov/31653696/), deglycase activity is a secondary effect that is derived from glyoxalase activity against MG that is in rapid equilibrium with reversible MG adducts. When explain the above results, the authors should be very careful whether the results are derived from deglycase activity of DJ-1 or from the secondary effects of glyoxalase activity of DJ-1.

We are fully aware of the controversy on this particular topic, and of the recent studies nicely showing that the glyoxalase activity accounts for the observed deglycase activity previously reported by G. Richarme (and indeed J. Dairou). In the context of the manuscript, we however focused on the observed inhibition of the protection against methylglyoxal, which is more or less independent from the mechanistic route. We nevertheless removed the term “glycation” as much as possible, using instead “protection against methylglyoxal”, and added a sentence on the recent studies showing that a glyoxalase activity accounts for the apparent deglycase activity.

(2) Fig. 4B

The authors showed the interesting results that sodium disulfide increases the thermal stability of DJ-1. This result is very interesting for me. At the same time, the authors reported that they were unable to observe a clean melting curve for the oxidized form of DJ-1 under their experimental DSF conditions. In addition, they could not observe stabilization of DJ-1 following incubation with hydrogen peroxide.

However, I had independently performed the thermal shift experiments of DJ-1, and confirmed that H2O2 stabilizes DJ-1 very clearly as has been reported previously (Ref 53). I think the best and the easiest way to detect the thermal stabilization of DJ-1 is using Protein Thermal Shift Assay Kit of Thermo Fisher Scientific.

Using that kit (Thermo Fisher Scientific, catalog # 4461146), H2O2-dependent stabilization of DJ-1 can be easily checked according to the manufacturer's protocol. The authors should perform such the experiment.

We have changed the Figure 4B to include additional data. Indeed, in the presence of DTT a similar Tm than the one reported by Wilson et al. (75°C) is obtained. The crucial role of DTT is mentioned in the main text, because it also slightly shifts the Tm of the persulfidated form, with a melting curve that is less well defined. A sentence was also added to briefly compare to thermal stabilities of the 

Round 2

Reviewer 1 Report

The authors have addressed my prior review comments.

Author Response

Thank you

Reviewer 2 Report

Thank you for the answers.

I could not understand the answer refering isatin. Maybe it is my fault, but I did not find any mention in the manuscript about it.

The statistics part of the manuscript is still insufficient. * symbols and p values are incomprehensible without the describtion of the statistical analysis. I higly recommend the authors to ask for guidance from a competent person.

Author Response

Thank you for the answers.

I could not understand the answer refering isatin. Maybe it is my fault, but I did not find any mention in the manuscript about it.

We just answered what we thought was a question outside the manuscript. Sorry for the misunderstanding

The statistics part of the manuscript is still insufficient. * symbols and p values are incomprehensible without the describtion of the statistical analysis. I higly recommend the authors to ask for guidance from a competent person.

We have now added a new sentence in the experimental part for this

Reviewer 3 Report

The revision of the manuscript "Persulfidation of DJ-1 : Mechanism and Consequences" by Galardon and colleagues has unfortunately barely improved the overall performance of this paper. The biochemical in vitro analysis seems sound and has improved by adding statistical analysis to some of their results. It could still be improved by statistically analyzing all results (e.g. Fig. 2A still not analyzed) and by also mentioning the method used for statistical analysis to give the reviewer and reader the possibility to evaluate whether appropriate methods were used. However, the main claim of the paper is a biological relevance of their findings by using an overexpression system in E.coli. The data presented (and especially the absence of data that I asked for) are at best not convincing and of very low scientific standard.  Therfore, I have to recommend to reject this manuscript.

1) The answer to my major remark 1 is OK. The authors elaborated a bit on the biological relevance of the chosen donors in the discussion.

2) Most results are now statistically analyzed, but the figure legends should include the statistical method used.

3) The major remark 3 is answered and OK.

4) The answer to my major remark4 is not satisfying. Using HEK 293T cells or HeLa cells or MEF is not more complicated or labor intensive than E. coli. Therefore, it is not beyond the scope of this study. In fact, at least for studying the wt DJ-1 it would save time and work because the analysis would be performed on endogenous protein and wouldn't require any transformation or induction. And as mentioned, since this has been done already by Filipovic and colleagues, results in E. coli are of little to no value to the scientific community.

5) My concerns I expressed in my remark 5 have not been addressed at all. The figure 5A is based on one single experiment. It has not been replicated, there are no additional data from independent experiments to perform a statistical analysis. And on top of that, the result is inconclusive. This is very bad scientific practice and not acceptable. Especially, because I pointed this out in the first review and requested: "authors should perform this experiment in multiple replications and quantify the bands and provide a summary with statistical analysis. As presented now, the authors have performed 2 experiments (100 and 150 μM Na2S2) once and draw conclusions from n=1."

The results presented in Figure S1' are from in vitro experiments with purified recombinant protein and can not be used for justification here. The entire point of testing in vivo is that it might be different from in vitro.

The Figure 5 A as it is right now should not be published.

6) Major remark 6 has been answered.

7) In the first review I wrote under major remark 7 that it is unacceptable to not show these important data and I thought it was just missing because the authors misjudged the importance of these data. But now that the authors refuse to show the data for the second time I have to conclude that it is intentional. To see the signal intensity of the C106S mutant and statistically compare it to the signal intensity of the wt DJ-1 is essential in this experiments. Otherwise the entire Figure 5 is worthless. The fact the authors did not include the data in the revised version makes me worry whether these data are hidden because they don't fit the authors hypothesis.

Author Response

We have now:

  • added the statistical method used as required (remark 2)
  • added the quantification of the bands (remark 5) and statistical analysis , with n=4 in Figure 5A (remark 5)
  • added the gels for E. coli lysate of wt vs C106S, as Figure S1'' (remark 7)

Round 3

Reviewer 2 Report

Dispite the clear recommendations, the authors did not complete the minimal statistical analysis of their data to meet the most basic requirements of scientific papers (usage of unpaired t-test can be used after normality test of the data; n=3 can not pass normality; unpaired t-test can not be used in case of more then two goups; compared groups are still not indicated properly, e.g. there is no "control" group in Fig. 2, Fig 5 is also confusing, etc.). Unfortunately, I can not recommend the acceptance of the manuscript.

Author Response

dear reviwer

  • We do not understant your comments: each set of data is compared with the controls, so two sets of data are compared at a time. Figure 2 contains a control ("% of the hydrolytic activity of purified DJ-1, determined as described in the experimental section"). Additionaly, It is not common in biochemistry to carry out statistical treatments on kinetic reaction.
  • Moreover, we used the statistical approach used by many in the field. For instance, in ref 30 of our manuscript (published in a journal with IF >25) , which is a seminal paper in our field, unpaired two-tailed t-test are used with n=3 throughout the manuscript.
  • Sincerely yours,
  •  

Reviewer 3 Report

- The authors analyze their results with an unpaired T-test, two-tailed I assume but it's not stated. This method is not appropriate to compare several groups. The results of Fig. 2B, Fig. 5A and Fig. 5C have to be checked for normal distribution of data points. If normally distributed the results have to be analyzed by ANOVA followed by an appropriate post-hoc multiple comparison method; if not normally distributed, the results have to be analyzed by a non parametric method followed by an appropriate post-hoc multiple comparison method. I suggest the authors seek guidance for statistical analysis of scientific data.

- Although this time the authors provide preliminary data (n=2, no quantification and no statistical comparison with the wt experiments) of the overexpression experiment with C106S DJ-1 in E.coli, they again fail to meet basic scientific standard. Obviously, the authors have the model and the means to run these experiments, so it's not understandable why they can not provide a complete data set with an appropriate number of replicate experiments, quantification of the signal and statistical analysis. This is the third review round and this should have been done (and was also requested) after the first round. To be clear, I am not doubting the result of the provided figures, however, without meeting basic scientific standard, these can not be published as evidence but are only preliminary data not fit for publication.

- Labeling of Fig. 5 needs correction. The legends states Fig. 5C, however the 'C' is missingin the figure.

Author Response

dear reviewer

  • We indeed used unpaired two-tailed t-*tests. We are making the figures even more clear in our new version of the manuscript
  • Caption of Figure 5 is now corrected